# Margination and adhesion dynamics of tumor cells in a real microvascular network

**Sitong Wang**, **Ting Ye***, **Guansheng Li**, **Xuejiao Zhang**, **Huixin Shi**

Department of Computational Mathematics, School of Mathematics, Jilin University, Changchun, China

* yeting@jlu.edu.cn

**Data Availability Statement:** All relevant data are within the manuscript and its Supporting information files.

**Funding:** Funding was received for this work to TY from Jilin Province Natural Science Foundation of

## Abstract

In tumor metastasis, the margination and adhesion of tumor cells are two critical and closely related steps, which may determine the destination where the tumor cells extravasate to. We performed a direct three-dimensional simulation on the behaviors of the tumor cells in a real microvascular network, by a hybrid method of the smoothed dissipative particle dynamics and immersed boundary method (SDPD-IBM). The tumor cells are found to adhere at the microvascular bifurcations more frequently, and there is a positive correlation between the adhesion of the tumor cells and the wall-directed force from the surrounding red blood cells (RBCs). The larger the wall-directed force is, the closer the tumor cells are marginated towards the wall, and the higher the probability of adhesion behavior happen is. A relatively low or high hematocrit can help to prevent the adhesion of tumor cells, and similarly, increasing the shear rate of blood flow can serve the same purpose. These results suggest that the tumor cells may be more likely to extravasate at the microvascular bifurcations if the blood flow is slow and the hematocrit is moderate.

## Author summary

Cancer is one of leading causes of death in the world, but unfortunately, the mechanism of tumor metastasis remains unclear so far. Intuitively, the tumor metastasis starts from a tumor cell migrating towards the vessel wall (namely margination), then adhering to the vessel wall (namely adhesion), and finally extravasating from where it adheres onto. Hence, it is important to investigate the margination and adhesion of tumor cells for understanding the tumor metastasis. We implemented a three-dimensional simulation to directly reproduce these two processes at a cellular scale, where the dynamic behaviors, such as deformation, aggregation and adhesion, of a lot of cells in a very complex microvascular network are taken into account. The results suggest that the tumor cells may be prone to adhere at the microvascular bifurcations with low shear rate and moderate hematocrit, because of a high wall-directed force from the surrounding RBCs. This implies that the tumor may be more likely to extravasate at the microvascular bifurcations if the blood flow is slow and the hematocrit is moderate. Our work may provide new insights into the cancer pathophysiology and its diagnosis and therapy.

China (http://kjt.jl.gov.cn), reference no.
20200201275JC. The funders had no role in study
design, data collection and analysis, decision to
publish, or preparation of the manuscript.

**Competing interests:** The authors have declared
that no competing interests exist.

## Introduction

Tumor metastasis is the major cause of cancer treatment failure, where tumor cells detach from the primary site, and then spread to distant organs or other parts of the body through blood and lymphatic systems, forming the secondary tumor [1–3]. The metastasis through the blood circulation is known as hematogenous metastasis, which was reported to cause 90% death of cancer [4], and the tumor cells entering into blood circulation are so-called the circulating tumor cells (CTCs). There are several critical steps for finishing a tumor metastasis, including intravasation, margination towards the vascular wall, adhesion onto the vascular wall, and extravasation from the circulation to a distant host organ [5]. We here pay our attention to the margination and adhesion of the CTCs in a real microvascular network, for providing a deep understanding of the tumor metastasis.

The margination and adhesion are two critical and closely related steps in tumor metastasis, and generally, the former step is often regarded as a prerequisite for the latter one. Both of behaviors were firstly observed on the leukocytes [6], instead of tumor cells, and hence, most of the subsequent studies have still focused on the leukocytes [7–14]. For example, Firrell et al. [7] and Abbitt et al. [9] suggested that the margination and adhesion of a leukocyte mainly occur at low flow rates based on in vivo and in vitro experiments. Jain et al. [11] demonstrated in the microfluidic experiments that the leukocytes have obvious margination within a range of hematocrit of 20–30%, and the lower or higher hematocrit may cause less margination. Fedosov et al. [12–14] also gave the similar dependence of leukocyte margination on the flow rate and hematocrit by numerical simulations. Some other types of cells, such as platelets [15–19] and malaria-infected RBCs [20, 21], have been found to have similar margination in the vessel. So far, there are few quantitative studies on the margination and adhesion of tumor cells. King et al. [22] pointed out that the soft CTCs are marginated more slowly than the rigid CTCs, because larger deformation of the soft CTCs caused by the collisions with surrounding RBCs can prolong the time that CTCs arrive at the vessel wall. However, some experiments [23–25] have shown that the soft CTCs are more likely to extravasate from the circulation, because the soft CTCs have a large contact surface with the vascular wall and thus have a higher possibility of adhesion than the rigid CTCs. Recently, some simulation work [26, 27] has also favored the latter conclusion.

A stochastic adhesive model was developed by Hammer and Apte [28], which has been widely used to theoretically and numerically investigate the adhesion of various cells, certainly including the CTCs [27]. The model proposed that the cell adhesion is attributed to the dynamic association and dissociation between the receptors on the cell and the ligands on the vessel wall. Obviously, if the receptors and ligands contact with each other frequently, the cell is more likely to adhere to the vessel wall. But, the contact frequency depends on a lot of parameters, such as the numbers of the receptors and ligands, the shear force of fluid flow, and local force from neighboring cells [5, 29–33]. Apart from these, some studies [34–36] have shown that the cell adhesion has a considerable dependence on the microvascular geometry. For example, Liu et al. [37] observed in vivo experiments that breast cancer cells preferentially extravasate from the vessel bifurcation. More recently, Hynes et al. [36] and Pepona et al. [38] studied the metastatic behavior within a complex vasculature via both experiments and simulations, and also confirmed that CTCs are prone to attach at the branch points. However, so far there are few studies on the tumor metastasis in a complex microvascular network at a cellular scale, at which the cell deformation, aggregation and adhesion should be taken into account simultaneously [39–41]. Experimental observations at this scale are limited by the reliability of measurements and the complexity of blood flow in microvascular networks [42, 43]; numerical modeling also poses great challenges due to

the poly-disperse feature of blood components and the network-like structure of microvessels [44, 45].

In the present work, we implement the direct three-dimensional modeling of tumor metastasis in a real and complex microvascular network extracted from the mesenteric microvessels of a rat. The deformation, aggregation and adhesion behaviors of cells are considered so as to deeper investigate the margination and adhesion dynamics of tumor cells. A hybrid method of smoothed dissipative particle dynamics and immersed boundary method (SDPD-IBM) is used as the main numerical methodology, which has been developed in our previous work [46]. It is found to be more suitable to solve the problems of fluid-structure interactions in a complex computational domain. The main aim of this study is to reveal the mechanism of tumor metastasis in a microvascular network, as well as the effects of hematocrit of RBCs and shear rate of blood flow.

A mesenteric tumor is taken for example in the present work, and it has been found in all age groups from infancy to the very elderly [47]. Fig 1A shows a segment of real microvessels in the rat mesentery [48]. One can consider human microvessels, and there are no large differences for numerical simulations, except that the vessel configurations are required to be reconstructed. In simulations, we choose a portion of this segment to construct microvascular network model by using the commercial software "SOLIDWORKS". This network is geometrically characterized by branching, merging and tortuous microvessels, and a small adjustment is made for getting only one outlet to suit the inflow and outflow boundary conditions used, as shown in Fig 1B. The length of the network is 260 $\mu$m, and the microvessels have circular cross section with the diameters ranging from 6.4 to 16 $\mu$m. Note that the diameter of each microvessel is calculated by Murray's law [49], a general law for a large number of transport networks, such as the vascular systems of animals, xylem in plants, and the respiratory system of

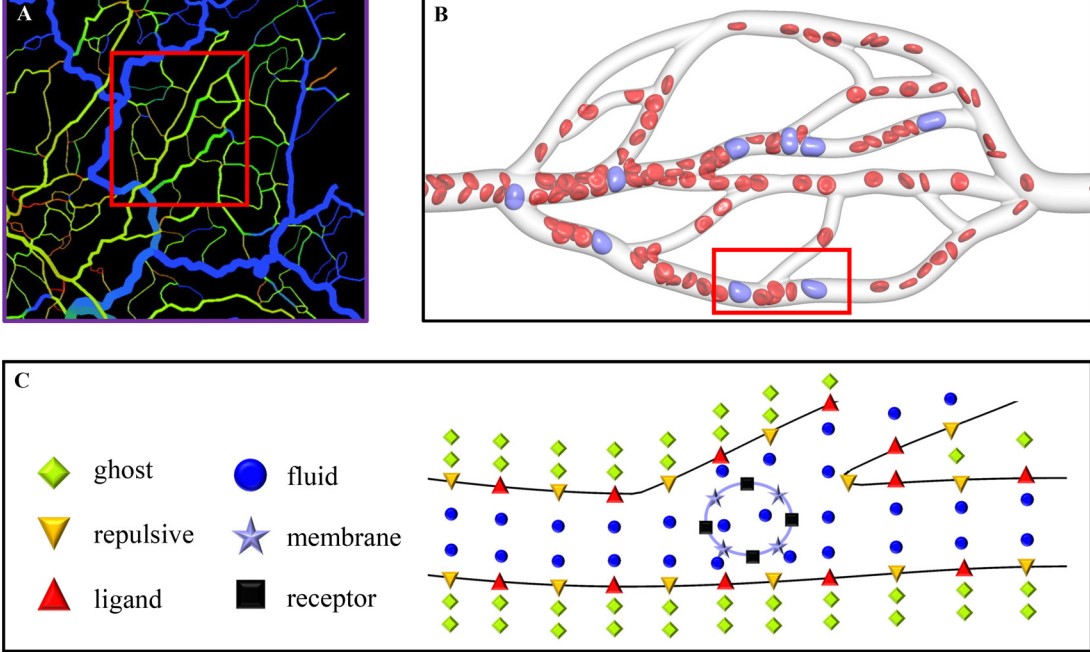

**Fig 1. Description of simulation problem.** (A) A segment of real microvessels in the rat mesentery [48], (B) the microvascular network extracted from the segment of microvessels, used as the simulation domain in the present work, and (C) the particle representation for a part of the microvascular network to show the particle discretization of the simulation domain.

insects. The microvascular network is filled only with plasma having no cells at the initial state. In order to make cells flow into the network, a cylindrical inlet with 120 $\mu$m in length is added at its left side, and it is arranged by a large number of cells, including RBCs and CTCs. An external force is imposed only in the inlet to generate an inflow, which pushes the cells into the network. Meanwhile, a cylindrical outlet with 20 $\mu$m in length is added at its right side for handling the cells that move out from the network. Hence, the total length of the simulation domain is actually 400 $\mu$m. The RBC is modeled to be a biconcave with a maximum diameter of 7.82 $\mu$m [50], while the CTC is spherical with the diameter of 9 $\mu$m. Due to the diversity of CTCs, their sizes are very different. In experiments, Gassmann *et al.* [51] observed that the diameter of a tumor cell in vivo is only about 5 $\mu$m, while Guo *et al.* [34] gave its diameter about 16 $\mu$m. In simulations, a moderate value of 8–10 $\mu$m is often chosen for saving computational cost [26, 27, 52]. The CTC's nucleus is also neglected for the same purpose, saving the computational cost, and this is one of limitations of our cell model. Fortunately, its effect on the CTC mechanics can be compensated to some extent by enhancing the CTC stiffness [53, 54]. The SDPD-IBM model, a particle-based method, is used as a numerical method in the present work, which discretizes the computational domain into a lot of particles. They are classified into six types in total, ghost particles, repulsive particles, fluid particles, membrane particles, receptor sites and ligand sites, respectively, as shown in Fig 1C. The ghost and repulsive particles are used to model the solid wall of the microvessels; the fluid particles are used to model the plasma and cell cytosol; the membrane particles are used to model the cell membrane; the receptor sites are scattered on the cell membrane, while the ligand sites are on the microvascular wall. The receptor and ligand sites are bonded for describing the CTC adhesion on the vascular wall. In the present work, there are 29704 ligand sites on the wall, 1176 receptor sites on the CTC's membrane, and no receptor sites on the RBC's membrane. It should be noted that these numbers are not same as the experimental values due to the limitation of computational resources. Hence, a coarse-grained technique is used, that is, a numerical receptor or ligand may represent several tens or hundreds of experimental receptors or ligands. The number of 29704 ligands is calculated by the fixed number density of 3 $\mu m^{-2}$, similar with that used in the work of Zhang *et al.* [55], while the number of 1176 receptors is similar with that used the work of Xiao *et al.* [27]. The ghost particles are 93116, the repulsive particles are 181810, and the fluid particles are 88771, respectively. Each RBC has 613 membrane particles, and each CTC has 1176 membrane particles equal to receptor sites for simplicity.

## Results

### Overview of CTC metastasis in the network

Fig 2 shows simulation snapshots of the CTCs in the microvascular network. At the initial state, there are 13 RBCs and one CTC placed into the microvascular inlet. They are driven into the microvascular network by an acceleration of $\bar{\boldsymbol{g}} = (9.79, 0, 0)$, which generates a mean flow velocity of $\bar{v}_m = 0.735$. Here, $\bar{\boldsymbol{g}}$ is scaled by $\varepsilon'/m'l'$, i.e., $\bar{\boldsymbol{g}} = \boldsymbol{g}/(\varepsilon'/m'l')$, $\bar{v}_m = v_m/\sqrt{\varepsilon'/m'}$, and $\bar{t} = t/l'\sqrt{m'/\varepsilon'}$. Two main phenomena are observed: i) the CTCs are deviated from the centerline of microvessels towards to the wall, known as the CTC margination; ii) the CTCs are more likely to be adhered at the bifurcation.

Take one of CTCs for example, the CTC-4 in Fig 2. At $\bar{t} = 429$, it arrives at the apex of a bifurcation, and a couple of bonds are formed on it. At $\bar{t} = 497.5$, it is ready to move into a branch, and more bonds are formed obviously. At $\bar{t} = 544.5$, it enters into the branch completely, and all the formed bonds are dissociated. Until $\bar{t} = 574$, it is still in the branch and there are no bonds formed any more. The other CTCs also present the similar adhesion

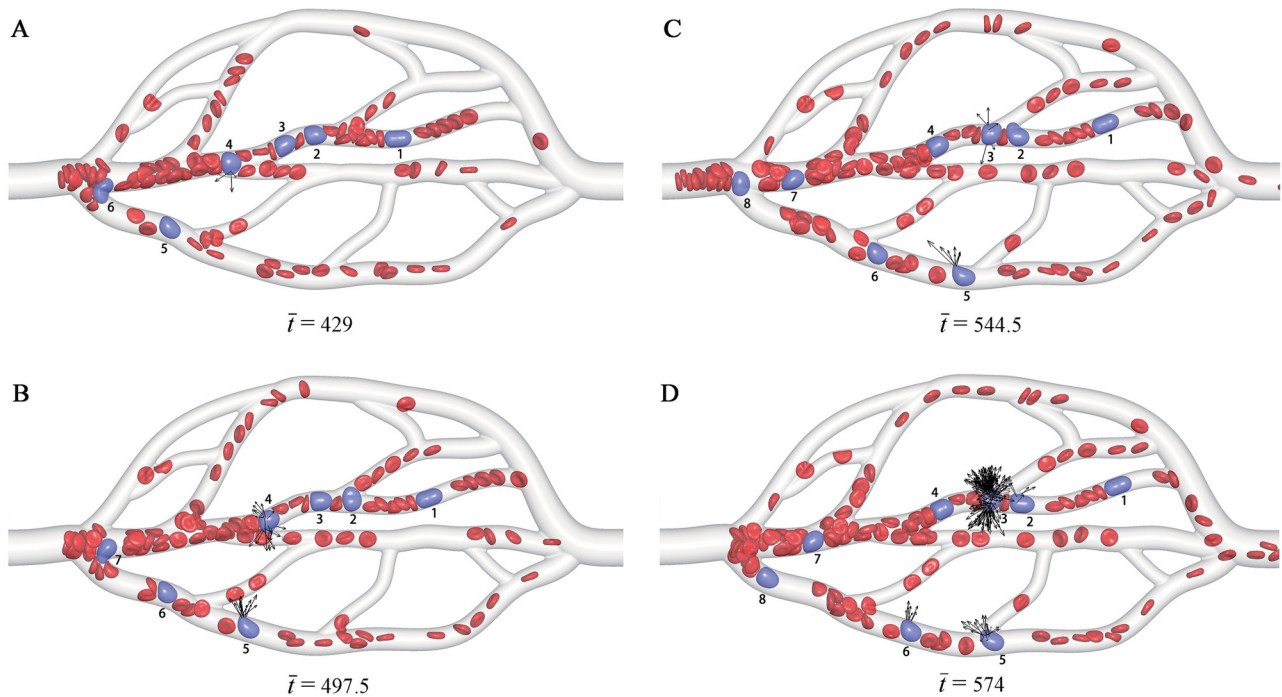

**Fig 2. Snapshots of the CTCs in the microvascular network at different time instants $\bar{t}$.** The RBCs are in red, while the CTCs are in purple and also labeled by Arabic numerals for identification. The black arrows show the adhesion force of the CTCs. For more snapshots, see S1 Video.

behavior, such as the CTC-1, 2, 3 and 5. This implies that the CTCs are more likely to be adhered at the bifurcation region, rather than in the branch.

To support this conclusion, a quantitative analysis is performed by examining the variations of bond number and retention time $\Delta\bar{t}$ with respect to the $x$−position of each CTC, as shown in Fig 3. It is found that the maximum number of bonds appears at a bifurcation region more often, and the CTC also stays for longer time at the bifurcation region. After the CTC enters into the branch completely, the bond number decreases to zero quickly, and it moves fast due to high velocity in a narrow branch. For example, when the CTC-6 arrives at about $\bar{X}_c = 19$ (near a bifurcation), about 4 bonds are formed in Fig 3A, and it stays for a time interval of $\Delta\bar{t} = 30$, as shown in Fig 3B. Similarly, when the CTC-4 moves to about $\bar{X}_c = 55$ (near another bifurcation), the number of bonds achieves a peak value of about 200 in Fig 3A, and it stays for a time interval of $\Delta\bar{t} = 17$, as shown in Fig 3B. After the CTC passes through the bifurcation, the bond number quickly decreases to zero while the velocity of CTC becoming faster. It is clear that this CTC does not experience a firm adhesion, and thus its adhesion is temporary. Generally, the formation of adhesive bonds at a bifurcation causes the longer time interval during which the CTCs stay at that position due to the adhesion force. In turn, the longer the CTC stays at a position, the larger the probability is to form a bond. As a result, it can be concluded that the tumor cells are indeed more likely to adhere at vascular bifurcations, and may further extravasate from the bifurcations into the tissue for finishing the tumor metastasis. This is consistent with the previous experiments [34], indicating that hemodynamic factors at vascular bifurcations play an important role in tumor metastasis.

Except the adhesion at a bifurcation, a CTC may also be adhered in a branch, e.g., the CTC-5, the CTC-6 in Fig 2, although this adhesion is not the mainstream. It is calculated that the distance between the CTC-5 centroid and the vessel wall ranges from 3.85 to 1.71 as the vessel

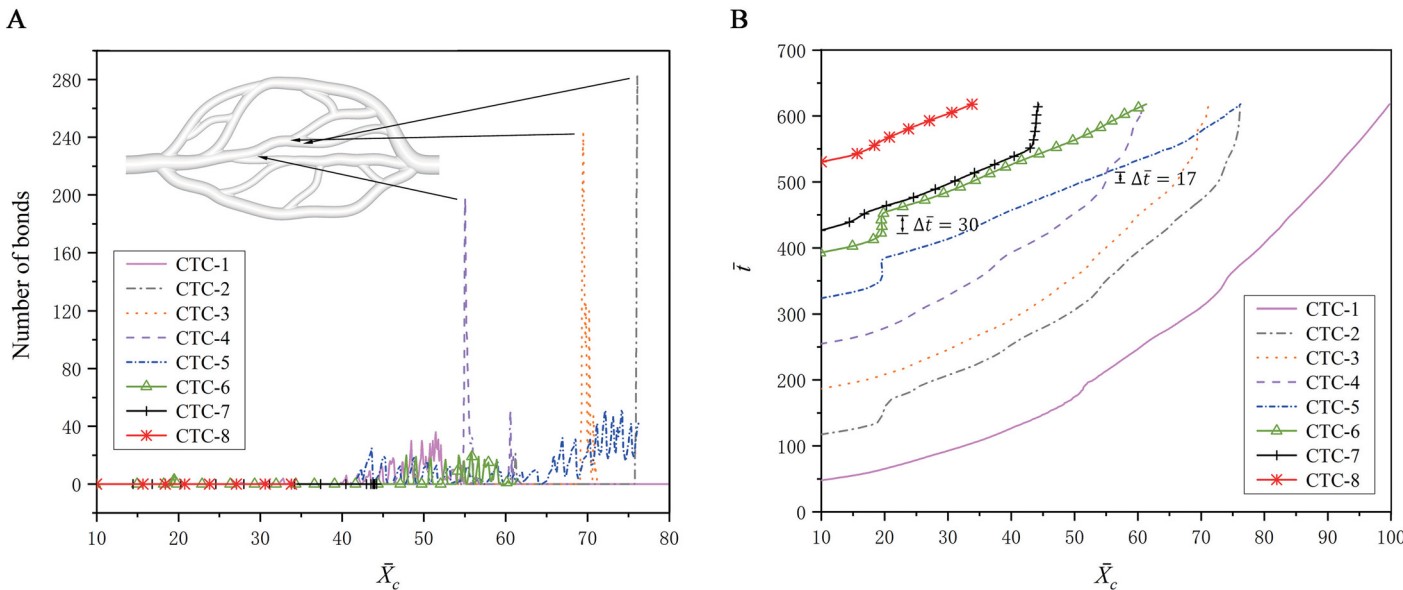

**Fig 3. Adhesion behavior of the CTCs in the microvascular network.** Variations of (A) the bond number and (B) the physical time with respect to the $x$-position $\bar{X}_c$.

diameter ranges from 7.7 to 5.57. That is, the CTC is deviated from the centerline of the vessel towards the wall, such that it arrives at the vessel wall completely. The similar margination behavior occurs with other CTCs, which provides more opportunities for the adhesion. The marginated CTC-5, CTC-6, CTC-7 are observed in Fig 2. Therefore, the margination is often regarded as a prerequisite for the adhesion.

## Margination of a CTC in a straight tube

To fully understand the margination of CTCs, we investigate the behavior of one CTC and 49 RBCs in a cylindrical tube with the diameter of $\bar{D}_v = 11$. The RBC hematocrit is calculated to be $H_{ct} = V_{RBC}/V_{tube} = 30\%$, corresponding to a normal level of RBCs in the real arteriole, where $V_{RBC}$ and $V_{tube}$ are the volumes of all RBCs and the tube. The accelerations of $\bar{g} = 19.15$ and 5.73 are applied to generate the fluid flow with the different mean velocities of $\bar{v}_m = 2.94$ and 0.88, respectively. Thus, the average shear rates are calculated to be $\bar{\dot{\gamma}} = \bar{v}_m/\bar{D}_v = 0.27$ and 0.08, corresponding to the physical values of 272 s$^{-1}$ and 81 s$^{-1}$. They are regarded as a high shear rate for the real arteriolar flow and a low shear rate for the venular flow in microcirculation, respectively [42, 56]. In addition, we use a periodic boundary condition in consideration of the simple straight cylindrical tube, such that a long simulation is allowed to be run. In simulations, the cells pass through the tube for 12 rounds, and thus the effect of the initial configurations of the cells can be negligible.

At the initial state, the CTC is placed at the tube centerline, while the RBCs are evenly distributed throughout the tube, as shown in Fig 4A. As the time elapses, the RBCs gradually migrate towards to the tube centerline, and the RBC-free layer is presented around the tube wall, as shown in Fig 4B. Meanwhile, the CTC is expelled away from the tube centerline, and an obvious margination behavior is observed in Fig 4C. This margination behavior is mainly attributed to the aggregation force acting on the CTC from the surrounding RBCs. Fig 5 shows the deviation of the CTC centroid from the tube centerline, and the radial aggregation force acting on the CTC from the RBCs. It is easily found that they are directly related. When the radial aggregation force is larger than zero, i.e., it is wall-directed, the deviation of the CTC

**Fig 4. Snapshots of the CTC margination in a straight tube.** These snapshots represent the motion states of RBCs and the CTC in the margination process. (A) The initial state, (B) RBCs move toward the center and (C) the CTC moves toward the wall.

centroid from the tube centerline continuously increases, that is, the CTC is pushed towards the vessel wall. In contrary, when this force is less than zero, i.e., it is center-directed, the CTC moves towards the tube centerline. Taking $\bar{\dot{\gamma}} = 0.08$ for example, as $\bar{X}_c \in [0, 70]$, the radial aggregation force is larger than zero, and thus the CTC continuously deviates from the tube centerline. As $\bar{X}_c \in [70, 120]$, the radial aggregation force is smaller than zero, and thus the CTC moves towards to the tube centerline.

Another phenomenon is also observed from Fig 5 that as the CTC gets closer to the wall, it is pushed back towards the tube centerline to some extent. Hence, the CTC cannot, in fact, approach to the vessel wall enough to exhibit the adhesion behavior, albeit the margination observed. We call this the incomplete margination, for example, the margination of CTC at $\bar{X}_c = 70$ under $\bar{\dot{\gamma}} = 0.08$, and the margination of CTC at $\bar{X}_c = 190$ under $\bar{\dot{\gamma}} = 0.27$. This is attributed to the local interaction between the CTC and the RBCs near the wall. As the CTC is close enough to the vessel wall, it is still surrounded by several RBCs. These RBCs provide a relatively strong center-directed aggregation force, because the distance between the CTC and RBCs is small enough at this moment. This force pushes the CTC back to the tube centerline to some extent.

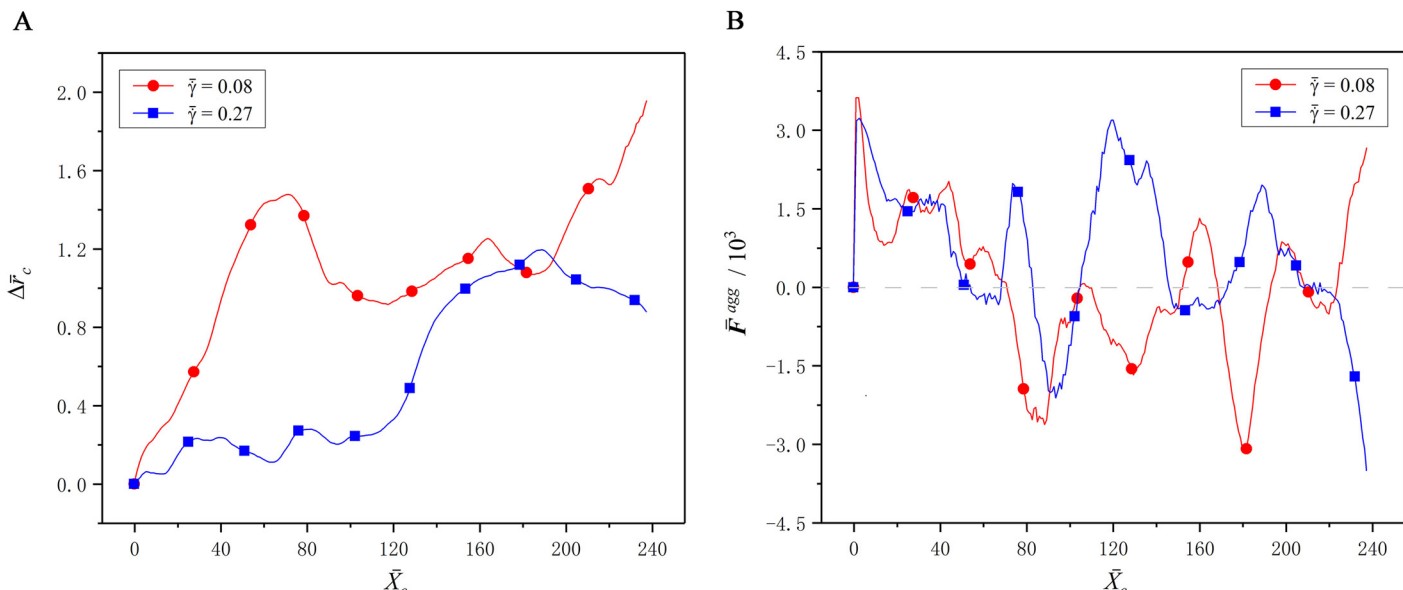

**Fig 5. Effect of the interaction force between the CTC and RBCs on margination behavior.** (A) The deviation $\Delta \bar{r}_c$ of the CTC centroid from the tube centerline at the shear rates of $\bar{\dot{\gamma}} = 0.08$ and $0.27$. (B) The radial aggregation force $\bar{F}^{agg}$ acting on the CTC from the RBCs, where the positive is the force directing to the tube wall, and the negative is the force directing to the tube centerline.

In addition, it is also found from Fig 5 that the wall-directed force at $\bar{\dot{\gamma}} = 0.08$ is less than that at $\bar{\dot{\gamma}} = 0.27$ on the whole, but the CTC margination at $\bar{\dot{\gamma}} = 0.08$ is more obvious. This is because the wall-directed force acts on the CTC much longer at the low shear rate than at the high shear rate. As a result, the CTC margination often occurs in the venules where the blood flows slowly although the wall-directed force is small, rather than the arterioles with a high shear rate. That is one of the reasons why the CTC adhesion is more likely to occur in the venules and rarely in the arterioles [34]. Certainly, if the vessel is narrow enough, such as the capillary in Fig 2D, the CTC can reach the microvascular wall completely, and its adhesion may happen.

## Adhesion of a deformable capsule on a plate

To fully understand the adhesion behavior and establish reliability of adhesion model, we investigate the adhesion behaviors of a deformable capsule, and compare the simulation results with the work of Zhang et al. [55]. The capsule is spherical with a radius of $R = 3.75\mu$m, which is placed into a cubic tube with the size $10R \times 6R \times 6R$. A Couette flow is generated by moving the top wall of the tube with a constant shear rate $\dot{\gamma} = 7000$ s$^{-1}$. The fluid density is $\rho = 10^3$ kg/m$^3$, and the viscosity is $\eta = 10^{-3}$ pa · s. The Reynolds number is $Re = 0.1$. The capillary number $Ca$ is set to be 0.005 and 0.015, such that the shear modulus of the capsule is $E_S = 5.25 \times 10^{-3}$ and $1.75 \times 10^{-3}$ N/m. The bending modulus is calculated by $E_B = 0.02R^2E_S$, and the dilation modulus is calculated by $E_D = 100E_S$. In the adhesion model, the equilibrium length is $\lambda = 50$ nm, and the reactive distance is $l_r = 375$ nm. The bond strength is determined by a dimensionless parameter $K_{SP} = E_A/\eta R\dot{\gamma} = 250$, that is, $E_A = 6.56 \times 10^{-3}$ N/m. The formation strength is $\sigma_f = 0.02E_A$, and the dissociation strength is $\sigma_d = 0.98E_A$. The unstressed formation and dissociation rates are determined by two dimensionless parameters, $K_f = k_f^0/\dot{\gamma} = 10$ and $K_d = k_d^0/\dot{\gamma} = 1.0$ (or 0.01), respectively. All of these parameters are set to be same as those used by Zhang et al. [55]

Fig 6 shows the three distinct adhesion states under $Re = 0.1$. At a high shear rate ($Ca = 0.015$) and a high dissociation rate ($K_d = 1.0$), the capsule detaches completely from the bottom wall and moves along the fluid flow. This state is called the detachment adhesion, as shown in Fig 6A. As the shear rate decreases ($Ca = 0.005$) but the dissociation rate remains

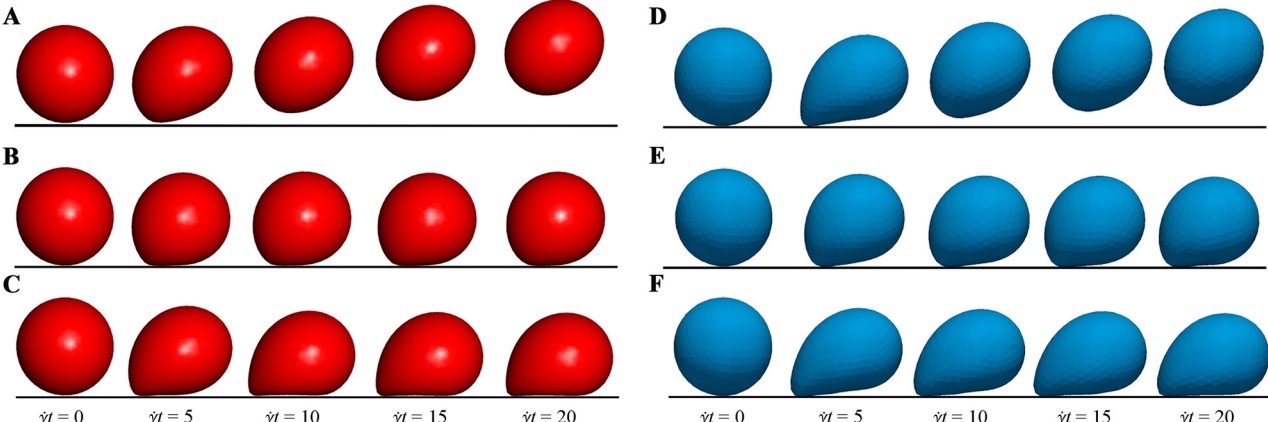

$\dot{\gamma}t = 0$ $\dot{\gamma}t = 5$ $\dot{\gamma}t = 10$ $\dot{\gamma}t = 15$ $\dot{\gamma}t = 20$ $\dot{\gamma}t = 0$ $\dot{\gamma}t = 5$ $\dot{\gamma}t = 10$ $\dot{\gamma}t = 15$ $\dot{\gamma}t = 20$

**Fig 6. Comparisons of the three adhesion states.** Comparisons of the three distinct adhesion states between our results (A-C) and those obtained by Zhang et al. [55] (D-F) under $Re = 0.1$. The detachment state (A and D) is obtained under $Ca = 0.015$ and $K_d = 1.0$; the rolling adhesion state (B and E) is obtained under $Ca = 0.005$ and $K_d = 1.0$; the firm adhesion state (C and F) is obtained under $Ca = 0.015$ and $K_d = 0.01$.

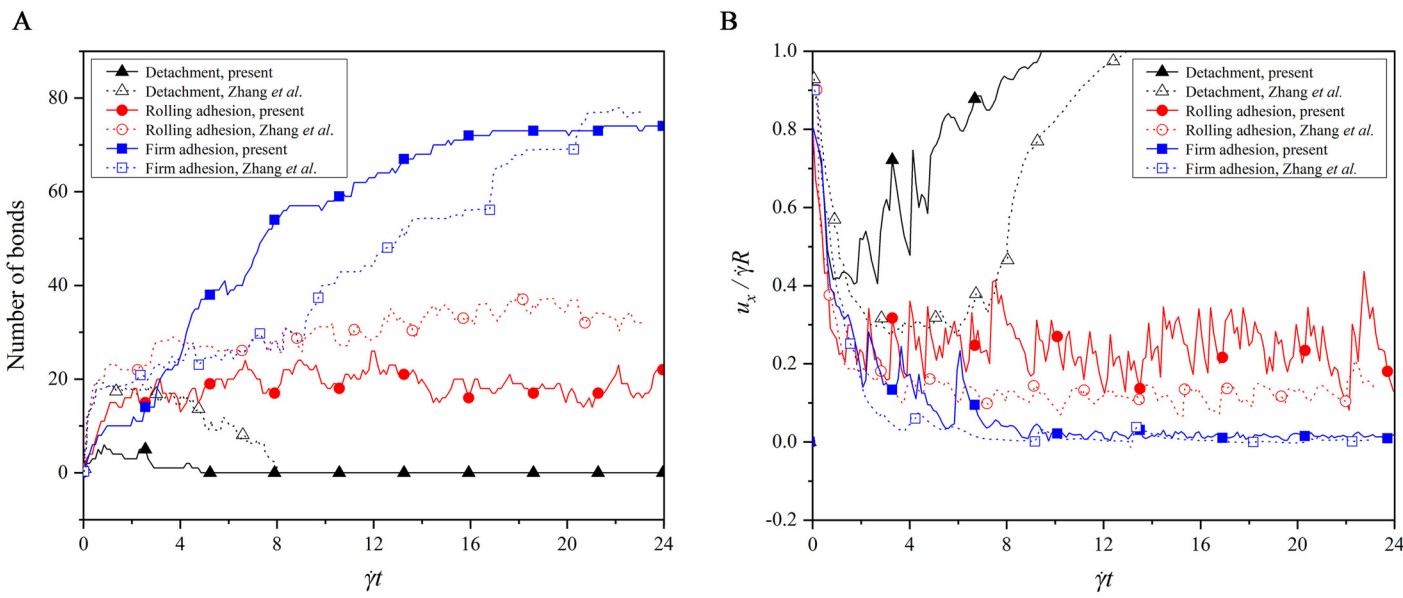

**Fig 7. Comparisons of (A) bond number and (B) translational velocity in the three adhesion states.**

unchanged ($K_d$ = 1.0), the capsule rolls on the bottom wall, and this state is called the rolling adhesion, as shown in Fig 6B. Finally, if the dissociation rate decreases ($K_d$ = 0.01) but the shear rate remains unchanged ($Ca$ = 0.015), the capsule is firmly adhered onto the wall and does not move any more. This state is called the firm adhesion, as shown in Fig 6C. It is clear that our results show the same three states as those of Zhang *et al.* [55].

Which adhesion state is present depends on the number of bonds formed between the receptors on the cell membrane and the ligands on the bottom wall. Fig 7 shows the time evolutions of the bond number and translational velocity in the three adhesion states. In the detachment adhesion, there are a few bonds formed at the beginning, but they are all dissociated quickly. This causes the capsule to exhibit a translational velocity that decreases first and then increases to move with fluid flow. In the firm adhesion, there are about 70 bonds formed at the steady state, which hold the capsule firmly on the bottom wall with the zero translational velocity. In the rolling adhesion, there are around 20 bonds that are kept for the whole simulation, leading to a translational velocity fluctuating around a constant value. To sum up, the rate of forming the new bonds is slower than that of dissociating the existing bonds in the detachment adhesion, but the opposite is for the firm adhesion. Both the rates are in equilibrium in the rolling adhesion.

In addition, a difference is noted that the bond number in our work is not exactly same as that in the work of Zhang *et al.* [55], as observed from Fig 7A. This difference is attributed to the difference of the deformation models used. A discrete particle model is used in our work, but Zhang *et al.* [55] adopted the Mooney-Rivlin model. Both models are approximated as linear at a small deformation, whereas at a large deformation, our cell has a larger deformation force than theirs if they undergoes the same deformation. That is, our model is strain hardening, while the model used by Zhang *et al.* [55] is strain softening. Therefore, the capsule in our work looks more rigid in each adhesion state, as shown in Fig 6, and thus they undergo the larger translational velocity in our work, as shown in Fig 7B. In the detachment and rolling adhesion, the rigid capsule is more likely to dissociate the bonds at a high dissociation rate $K_d$ = 1.0, such that the bond number in our work is less than that in the work of Zhang *et al.* [55]

In the firm adhesion, however, the rigid capsule is observed to have the larger contact surface with the bottom wall when its rear is firmly adhered on the bottom wall with $K_d = 0.01$, leading to more bonds formed in our work than that given by Zhang *et al.* [55] It can be found from this analysis that the cell deformation, cell adhesion and shear fluid are entangled and interact with each other.

## Adhesion of a CTC in a bifurcation

As mentioned above, the CTC adhesion happens more often in a bifurcation region, and hence we here focus on its behaviors in a 'single' bifurcation for a deeper understanding. The bifurcation is directly extracted from the microvascular network in Fig 1B, and the effects of RBC hematocrit and shear rate of fluid on the CTC adhesion are investigated in this section.

Fig 8 shows the effect of the RBC hematocrit, which is adjusted to be $H_{ct}$ = 10–40% by varying the number of RBCs in the microvascular inlet. The fluid flow has a mean velocity of $\bar{v}_m$ of 0.735 in all these cases. It is found that the bond number at $H_{ct} \geq 30\%$ is obviously greater than that at $H_{ct} < 30\%$. Its peak value is larger than 11 for $H_{ct} \geq 30\%$, but between 4 to 7 for $H_{ct} < 30\%$, as shown in Fig 8B. This implies that the CTC has the strongest adhesion at $H_{ct}$ = 30%, and this is also supported by the radial aggregation force that achieves the maximum value at $H_{ct}$ = 30%, as shown in Fig 8C. This result seems to contradict the expectation that an increase of $H_{ct}$ can enhance the CTC adhesion because of the increasing repulsion from the more RBCs. In fact, this expectation is not true for all the hematocrits. From a low hematocrit, such as $H_{ct}$ = 10–25%, the radial aggregation force indeed increases as $H_{ct}$ increases to 30%, as shown in Fig 8C,

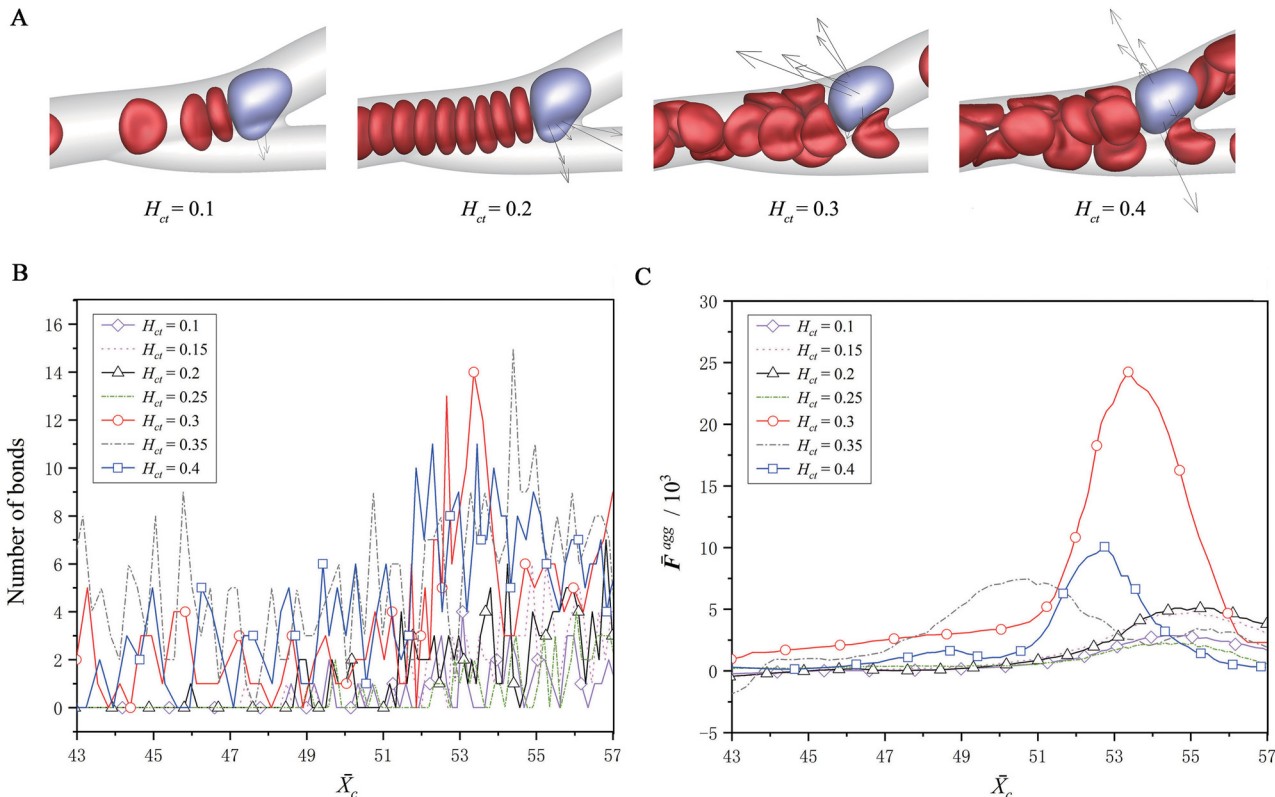

**Fig 8. Effect of the RBC hematocrit on the CTC adhesion.** (A) The snapshots of the CTC at about $\bar{X}_c = 54$, the variations of (B) the bond number and (C) the radial aggregation force from the RBCs, at the RBC hematocrits of $H_{ct}$ = 10–40%.

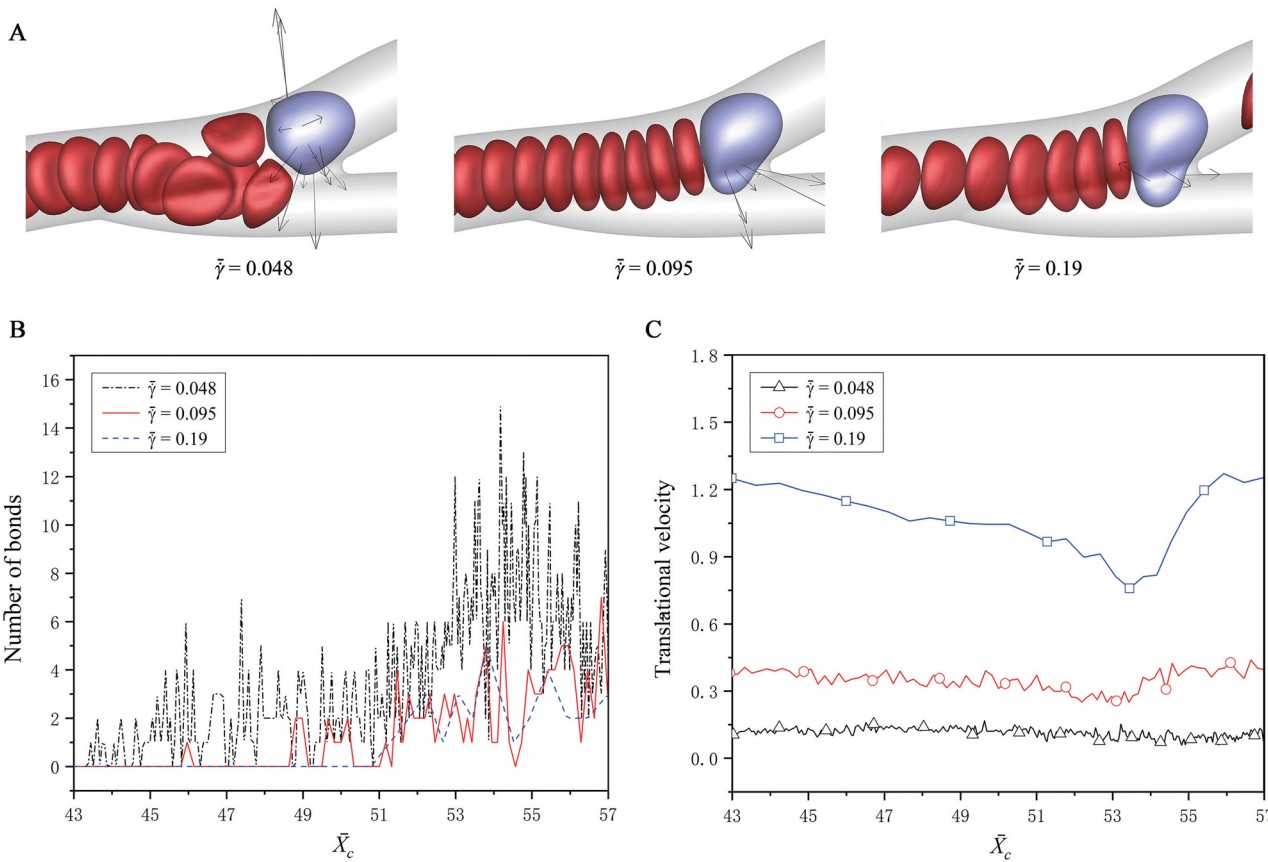

**Fig 9. Effect of the shear rate of fluid on the CTC adhesion.** (A) The snapshots of the CTC at about $\bar{X}_c = 54$, the variations of (B) the bond number, and (C) the translational velocity, at the shear rates of $\bar{\bar{\gamma}} = 0.048$, 0.095 and 0.19.

which in turn promotes the CTC adhesion. This was also observed in the work of Xiao *et al.* [27], where $H_{ct}$ ranging from 10 to 30% was considered. However, when the hematocrit varies from a normal value to a higher value, for example from $H_{ct} = 30$ to 40%, the radial aggregation force does not increase but instead decreases, and correspondingly the CTC adhesion becomes weak. This is because the RBC-free layer is narrowed down subject to the more RBCs at a high hematocrit. Thus, the CTC is more likely to be surrounded by those RBCs, as shown in Fig 8A, and the outer RBCs give the CTC a center-directed aggregation force, counteracting the wall-directed aggregation force acting on the CTC. This was observed for the leukocytes in the work of Fedosov *et al.* [12] To sum up, the low hematocrit cannot provide the enough wall-directed aggregation force to promote the CTC adhesion, because of the small RBC number. The high hematocrit cannot also promote the CTC adhesion because the wall-directed aggregation force is offset by the more RBCs surrounding the CTC.

Fig 9 shows the effect of the shear rate of fluid flow, which is adjusted to be $\bar{\bar{\gamma}} = 0.048$, 0.095 and 0.19 by varying the externally-applied acceleration. The fluid flow has the mean velocity of $\bar{v}_m = 0.37$, 0.735, 1.47, respectively, and the RBC hematocrit is fixed as $H_{ct} = 20\%$. A conclusion is drawn that the CTC adhesion becomes weak as the shear rate increases, which was also confirmed in the recent simulation work of Dabagh *et al.* [26]. For example, at about $\bar{X}_c = 54$, there are 13, 6 and 3 bonds formed for $\bar{\bar{\gamma}} = 0.048$, 0.095 and 0.19, respectively. This is directly attributed to the increase of the translational velocity of the CTC with increasing the

shear rate, as shown in Fig 9C. At a low shear rate, $\bar{\dot{\gamma}} = 0.048$, the cells move slow so that the RBCs is aggregated obviously, as shown in Fig 9A, leading to a strong wall-directed aggregation force to promote the CTC adhesion. At a high shear rate, $\bar{\dot{\gamma}} = 0.19$, the RBCs are distributed more dispersedly, and the less wall-directed aggregation force is provided for the CTC adhesion. Moreover, it is worth mentioning that the CTC obtains the maximum bond number at the bifurcation (about $\bar{X}_c = 54$), regardless of discussing the effects of hematocrit or shear rate. This again confirms that the CTC is more likely to be adhered at the position of a bifurcation. The similar phenomena have been observed in experiments. Liu *et al.* [37] found that MDA-MB-231 human breast cancer cells adhere at the microvascular bifurcations, and then extravasate from the blood vessel, by an in vivo experiment. Guo *et al.* [34] also observed that the tumor cells may more often adhere at the bifurcated regions of vessels, as well as small venules with relatively low shear rates.

## Discussion

A direct three-dimensional simulation of tumor cells in a complex microvascular network was carried out to understand the metastasis of tumor cells as realistic as possible. The microvascular network was constructed from a real mesenteric vasculature of a rat, and comprised of bifurcating, merging and winding vessels. The cells include the red blood cells (RBCs) and circulating tumor cells (CTCs), without the platelets due to their small portion in blood. Three mechanical behaviors of cells, the deformation, aggregation and adhesion, were taken into account in the present work.

The margination and adhesion of tumor cells were mainly investigated, as well as the effects of the RBC hematocrit and flow shear rate. The simulation results showed that the tumor cells adhere more frequently at the microvascular bifurcations, and there is a positive correlation between the adhesion and wall-directed force from the surrounding RBCs. The larger the wall-directed force is, the closer the tumor cells are marginated towards the wall, and the higher the probability of adhesion behavior happen is. At a low hematocrit, e.g., 10%, the RBCs cannot provide the enough wall-directed force, because of the small RBC number, while at a high hematocrit, e.g., 40%, the wall-directed force is offset by the outer RBCs surrounding the CTC, because more RBCs exist in the vessels. Hence, a moderate hematocrit of 30% that is close to the normal level of hematocrit in a real arteriole, is found to have the largest wall-directed force, and have the largest bond number. Moreover, it is also found that the CTC margination and adhesion are enhanced as the shear rate decreases. At a low shear rate, the cells are transported slowly and aggregated easily, and the wall-directed force lasts long on a CTC. Hence, the CTC margination is more obvious at the low shear rate than at the high shear rate, and the bond number is larger, promoting the CTC adhesion.

To sum up, the present work suggests that the tumor cells may be more likely to adhere at the microvascular bifurcations with low shear rate and moderate hematocrit, because of a high wall-directed force from the surrounding RBCs. This may help to predict the location where tumor cells extravasate from the circulation, and further give new insights into the cancer pathophysiology and its diagnosis and therapy.

## Methods

### SDPD-IBM model for fluid flow

The Navier-Stokes (N-S) equations are adopted to govern the motion of the plasma and cytosol, given by [57]

$$\nabla \cdot \boldsymbol{v} = 0, \tag{1}$$

$$\rho \frac{\mathrm{d}\mathbf{v}}{\mathrm{d}t} = -\nabla P + \eta \nabla^2 \mathbf{v} + \rho \mathbf{g} + \mathbf{f}, \tag{2}$$

where $\rho$, $\mathbf{v}$ and $t$ are the density, velocity and time, respectively; $P$ is the pressure field, $\eta$ is the shear viscosity, $\mathbf{g}$ is an acceleration externally applied to drive the fluid flow, and $\mathbf{f}$ is a singular force from the cell membrane. The interaction between the cell and fluid is modeled by the immersed boundary method (IBM). The action on the fluid from the cell is expressed as

$$\mathbf{f}(\mathbf{x}, t) = \int_\Gamma \mathbf{f}^{cell}(r, s, t) \delta(\mathbf{x} - \mathbf{X}(r, s, t)) \mathrm{d}r \mathrm{d}s, \tag{3}$$

while the action on the cell membrane from the fluid is shown by the membrane evolution, i.e.,

$$\frac{\mathrm{d}\mathbf{X}}{\mathrm{d}t} = \int_\Omega \mathbf{v}(\mathbf{x}, t) \delta(\mathbf{x} - \mathbf{X}(r, s, t)) \mathrm{d}\mathbf{x}, \tag{4}$$

where $\mathbf{f}^{cell}$ is the force acting on the cell membrane due to the cell deformation, aggregation and adhesion, etc. $\mathbf{x}$ is the position of the fluid in the Eulerian system, $\mathbf{X}$ is the position of the cell membrane in the Lagrangian system, and $(r, s)$ denotes a curvilinear coordinate on the cell membrane to label a Lagrangian point. $\Gamma$ is the surface domain occupied by the cell membrane, and $\Omega$ is the computational domain. It should be here pointed out that both the cytosol and plasma are assumed to be a single type of fluid with the same physical properties, and they are incompressible and isothermal Newtonian fluid. The whole blood consisting of the plasma and the cells behaves as non-Newtonian fluid, because the cells are incorporated by the singular force $\mathbf{f}$, and thus the blood viscosity is described to depend on the cell hematocrit [58].

The SDPD-IBM method is used to discretize Eqs 2 and 4, and it has been already developed well in our previous work [46, 59], and here we only outline its framework for completeness. The fluid particles are evolved by

$$\mathrm{d}\mathbf{x}_i = \mathbf{v}_i \mathrm{d}t, \tag{5}$$

$$m\mathrm{d}\mathbf{v}_i = \mathbf{F}_C \mathrm{d}t + \mathbf{F}_D \mathrm{d}t + \mathbf{F}_R + \mathbf{F}_G \mathrm{d}t + \mathbf{F}_B \mathrm{d}t, \tag{6}$$

where $m$, $\mathbf{v}_i$ and $\mathbf{x}_i$ are the mass, velocity and position of the fluid particle $i$, respectively. The force $\mathbf{F}_C$ is the conservative force describing the compressibility of fluid. $\mathbf{F}_D$ is the dissipative force describing the viscosity of fluid. $\mathbf{F}_R$ is the random force to keep a constant Boltzmann temperature of fluid. See S1 Appendix for the detailed formulations of these three forces. $\mathbf{F}_G$ is an externally-applied force to drive the fluid flow. $\mathbf{F}_B$ is the force from the cell membrane, defined as

$$\mathbf{F}_B = \sum_k \beta_{ik} (\mathbf{F}_k^{def} + \mathbf{F}_k^{agg} + \mathbf{F}_k^{adh}), \tag{7}$$

where $\beta_{ik}$ is the weighted coefficient determined by the kernel function of SDPD, and $\mathbf{F}_k^{def}$, $\mathbf{F}_k^{agg}$ and $\mathbf{F}_k^{adh}$ are the membrane forces due to the cell deformation, aggregation and adhesion, respectively. The membrane particles are evolved by

$$\mathrm{d}\mathbf{X}_k = \sum_i \beta_{ik} \mathbf{v}_i \mathrm{d}t, \tag{8}$$

where $\mathbf{X}_k$ is the position of the membrane particle $k$. The ghost and repulsive particles are

stationary during the whole simulation, as well as the ligand sites. However, the positions of the receptor sites are evolved with the membrane particles, because the membrane particles are assumed to be same as the receptor sites.

## Discrete elastic model for cell deformation

To characterize the cell deformation, the cell membrane is modeled as a triangular network by connecting all membrane particles. The edges of each triangle are modeled as springs to describe the membrane elasticity, and the angles of any two neighboring triangles are used to describe the membrane bending. Moreover, the membrane area is restrained to be varied in 3%, due to its strong membrane elasticity [60]. The cell volume is also restrained to be varied in 3%, because the material exchange is often not considered in the previous work [61–63]. As a result, the total deformation potential energy $U^{def}$ of the triangular network is given by [64, 65]

$$U^{def} = U_s + U_b + U_a + U_v, \tag{9}$$

and the deformation force is calculated as

$$\boldsymbol{F}_k^{def} = -\frac{\partial U^{def}}{\partial \boldsymbol{X}_k}, \tag{10}$$

where $U_s$, $U_b$, $U_a$ and $U_v$ denote the in-plane energy, bending energy, area-restraint energy and volume-restraint energy, respectively. See S1 Appendix for more details about these four energy.

The deformation model is associated with there important macro parameters: the shear modulus $E_S$, the bending modulus $E_B$, and the dilation modulus $E_D$ [64],

$$E_S = \frac{\sqrt{3}k_B T}{4p_j l_j^0} \left[ \frac{s_0}{2(1-s_0)^3} + \frac{1}{2(1-s_0)^2} + 3s_0 - \frac{1}{2} \right], \tag{11}$$

$$E_B = \frac{2}{\sqrt{3}} K_B, \tag{12}$$

$$E_D = 2E_S + K_{AG} + K_{AL}, \tag{13}$$

where $k_B T$ and $p_j$ are the Boltzmann temperature and persistence length, respectively. $l_j^0$ is the length of the triangular edge $j$ at the stress-free state, $s_0 = l_j^0/l_j^d$, $l_j^d$ is the maximum length of the triangular edge $j$. $K_B$, $K_{AG}$ and $K_{AL}$ are the bending coefficient, global area restraint constant and local area restraint constant, respectively.

## Morse potential model for cell aggregation

To describe the cell-cell interaction, the Morse potential model proposed by Liu *et al.* [66] is used, in which the total aggregation energy between the cells is approximated as [67],

$$U^{agg} = \sum_{m=1}^{N_t} \varphi(r_{mm'})(\boldsymbol{n}_m \cdot \boldsymbol{k}_m)(\boldsymbol{n}_{m'} \cdot \boldsymbol{k}_{m'})A_m, \tag{14}$$

and the aggregation force is given by,

$$\boldsymbol{F}_k^{agg} = -\frac{\partial U^{agg}}{\partial \boldsymbol{X}_k}. \tag{15}$$

where $N_t$ is the number of triangles, and $A_m$ is the area of the triangle $m$. $\varphi(r_{mm'})$ is the Morse potential energy between two facing plane elements of the separate cells, defined as [66]

$$\varphi(r_{mm'}) = E_I[e^{2\beta(r_0 - r_{mm'})} - 2e^{\beta(r_0 - r_{mm'})}], \tag{16}$$

where $r_{mm'}$ is the local distance between the two elements, $E_I$ is the strength of surface energy, $r_0$ is the zero-force distance, and $\beta$ is a scaling factor. The term of $(\boldsymbol{n}_m \cdot \boldsymbol{k}_m)(\boldsymbol{n}_{m'} \cdot \boldsymbol{k}_{m'})$ is added to consider the effect of curved elements instead of plane elements by the DLVO theory [68], where $\boldsymbol{n}$ is the outward unit normal vector of a triangle, and $\boldsymbol{k}$ is the unit vector parallel to the line connecting the centers of two interacting cells. This model behaves as a weak attractive force at a far distance ($r_{mm'} > r_0$), whereas at a near distance ($r_{mm'} < r_0$) it behaves as a strong repulsive force.

## Stochastic binding model for cell adhesion

To characterize the CTC adhesion, the stochastic binding model proposed by Hammer and Apte [28] is used, in which an adhesion potential energy is given by,

$$U^{adh} = \frac{1}{2}\sum_{m=1}^{N_b} E_A(x_m - \lambda)^2, \tag{17}$$

and the adhesion force is thus calculated by

$$\boldsymbol{F}_k^{adh} = -\frac{\partial U^{adh}}{\partial \boldsymbol{X}_k}, \tag{18}$$

where $E_A$ is the adhesion strength, $N_b$ is the number of formed bonds, $x_m$ is the bond length, and $\lambda$ is an equilibrium distance. When a new bond is formed, it contributes to the adhesion potential energy. On the contrary, when an existing bond is dissociated, the corresponding contribution should be eliminated.

The formation and dissociation of a bond are stochastic processes, controlled by a formation probability $p_f$ and a dissociation probability $p_d$, respectively [28],

$$p_f = 1 - \exp(-k_f \Delta t), \tag{19}$$

$$p_d = 1 - \exp(-k_d \Delta t), \tag{20}$$

where $\Delta t$ is the time step. $k_f$ and $k_d$ are the formation and dissociation rates of a bond, defined as

$$k_f = k_f^0 \exp\left(-\frac{\sigma_f(x_m - \lambda)^2}{2k_B T}\right), \tag{21}$$

and

$$k_d = k_d^0 \exp\left(\frac{\sigma_d(x_m - \lambda)^2}{2k_B T}\right), \tag{22}$$

where $k_f^0$, $k_d^0$, $\sigma_f$ and $\sigma_d$ are four constants, and $k_B T$ is Boltzmann temperature. $k_f^0$ and $k_d^0$ are the unstressed formation and dissociation rates of bonds at the equilibrium distance $\lambda$; $\sigma_f$ and $\sigma_d$ are the formation and dissociation strengths within a given reactive distance $l_r$. If a bond is

formed, the following two conditions must be satisfied,

$$x_m < l_r, \text{ and } p_f > \xi_1, \tag{23}$$

where $\xi_1$ is a random number with uniform distribution in [0, 1]. If an existing bond is dissociated, one of the following conditions needs to be satisfied,

$$x_m \geq l_r, \text{ or } p_d > \xi_2, \tag{24}$$

where $\xi_2$ is another random number similar to $\xi_1$.

## Numerical details

There are two types of boundary conditions for Eqs 5–8, the solid boundary condition, and inflow/outflow boundary condition. The former one is applied on the microvascular wall for two purposes. One is to improve the accuracy near the microvascular wall by introducing the ghost particles. The other is to avoid the fluid and membrane particles to penetrating the microvascular wall by introducing the repulsive particles. More details about this boundary condition can be found in the work of Liu *et al*. [69] The inflow/outflow boundary condition is applied in the inlet/outlet of microvascular network, also for two purposes. One is to keep cells continuously moving into the microvascular network, and the other is to settle down the fluid and membrane particles that move out from the microvascular network. This boundary condition is one of advantages of our model, which not only ensures that the system including cells and fluid have reached steady states when they flow into the microvascular network, but also guarantees that the mass and momentum of the system are conserved. More details about this boundary condition can be found in our previous work [59].

A non-dimensional procedure is carried out, by choosing the cut-off radius, particle mass and Boltzmann temperature as the characteristic length $l'$, mass $m'$ and energy $\epsilon'$. They are set

**Table 1. Physical quantities and their characteristic quantities.**

| Physical quantities | Physical values | Characteristic quantities | Simulation values |
|---|---|---|---|
| Fluid density ($\rho$) | $1.0 \times 10^3 \text{kg/m}^3$ | $m'/l'^3$ | 8 |
| Shear viscosity ($\eta$) | $1.0 \times 10^{-4} \text{pa} \cdot \text{s}$ | $\sqrt{m'\epsilon'}/l'^2$ | 197 |
| Temperature ($T$) | 300K | $\epsilon'$ | 1 |
| CTC diameter ($D_t$) | $9.0\mu m$ [26] | $l'$ | 4.5 |
| RBC diameter ($D_r$) | $7.82\mu m$ [50] | $l'$ | 3.91 |
| Shear modulus of CTC ($E_S$) | $1.0 \times 10^{-6}$N/m [71] | $\epsilon'/l'^2$ | $9.66 \times 10^2$ |
| Shear modulus of RBC ($E_S$) | $6.0 \times 10^{-6}$N/m [72] | $\epsilon'/l'^2$ | $5.794 \times 10^3$ |
| Bending modulus of CTC ($E_B$) | $1.35 \times 10^{-19}$J [73] | $\epsilon'$ | 33 |
| Bending modulus of RBC ($E_B$) | $2.07 \times 10^{-19}$J [72] | $\epsilon'$ | 50 |
| Dilation modulus of CTC ($E_D$) | $1.16 \times 10^{-4}$N/m$^\dagger$ | $\epsilon'/l'^2$ | $1.12 \times 10^5$ |
| Dilation modulus of RBC ($E_D$) | $1.26 \times 10^{-4}$N/m$^\dagger$ | $\epsilon'/l'^2$ | $1.22 \times 10^5$ |
| Equilibrium length ($\lambda$) | $0.2\mu m$ [55] | $l'$ | 0.1 |
| Reactive distance ($l_r$) | $1.0\mu m$ [55] | $l'$ | 0.5 |
| Adhesion strength ($E_A$) | $3.0 \times 10^{-5}$N/m [55] | $\epsilon'/l'^2$ | $2.9 \times 10^4$ |
| Formation strength ($\sigma_f$) | $6.0 \times 10^{-7}$N/m [55] | $\epsilon'/l'^2$ | $5.8 \times 10^2$ |
| Dissociation strength ($\sigma_d$) | $7.54 \times 10^{-7}$N/m [55] | $\epsilon'/l'^2$ | $7.28 \times 10^2$ |
| Unstressed formation rate ($k_f^0$) | $1.205 \times 10^6 \text{s}^{-1}$ [55] | $\sqrt{\epsilon'/m'}/l'$ | $1.205 \times 10^3$ |
| Unstressed dissociation rate ($k_d^0$) | $3.55 \times 10^5 \text{s}^{-1}$ [55] | $\sqrt{\epsilon'/m'}/l'$ | $3.55 \times 10^2$ |

$^\dagger$ The dilation modulus in simulations is, in general, smaller than real values to save computational cost, as long as it can guarantee that the variation of surface area of a RBC or tumor cell is less than 3%.

as: $l' = 2\ \mu$m, $m' = 1.0 \times 10^{-15}$ kg, and $\epsilon' = 4.142 \times 10^{-21}$ J, respectively. The other physical parameters can be scaled by these three parameters; for example, the shear modulus $E_S$ is scaled by $\varepsilon'/l'^2$, and the bending modulus $E_B$ is directly scaled by $\varepsilon'$. Table 1 lists the physical parameters and their corresponding simulation values used in the present work. In the present work, we add an overline on the physical quantity ($\bar{\ }$) to stand for its dimensionless form. After that, the velocity-Verlet algorithm [70] is employed to numerically solve SDPD-IBM model in Eqs 5–8, because of its advantages of high computational efficiency. This algorithm is parallelized by the technique of message passing interface (MPI) to reduce the computational time. The detailed procedure about this algorithm can be found in our previous work [59]. The computational cost is about 51,600 core · hour for a typical simulation case.

## Supporting information

**S1 Video. Metastasis of tumor cells in the microvascular network.**
(MP4)

**S1 Appendix. Mathematical models and numerical methods.**
(PDF)

## Acknowledgments

The authors would like to thank School of Mathematics at Jilin University of China for all the supports, especially the valuable discussions with the members and the access to computational resources.

## Author Contributions

**Conceptualization:** Sitong Wang, Ting Ye.

**Data curation:** Sitong Wang, Ting Ye.

**Formal analysis:** Sitong Wang, Ting Ye.

**Investigation:** Sitong Wang.

**Methodology:** Sitong Wang, Ting Ye, Guansheng Li.

**Software:** Sitong Wang, Ting Ye, Guansheng Li.

**Supervision:** Ting Ye.

**Validation:** Ting Ye, Huixin Shi.

**Visualization:** Sitong Wang, Ting Ye, Xuejiao Zhang, Huixin Shi.

**Writing – original draft:** Sitong Wang, Ting Ye.

**Writing – review & editing:** Sitong Wang, Ting Ye.

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
