## [Decision Letter · Decision Letter 0]

10 Nov 2020

Dear Dr. Ye,

Thank you very much for submitting your manuscript "Margination and adhesion dynamics of tumor cells in a real microvascular network" for consideration at PLOS Computational Biology.

As with all papers reviewed by the journal, your manuscript was reviewed by members of the editorial board and by several independent reviewers. In light of the reviews (below this email), we would like to invite the resubmission of a significantly-revised version that takes into account the reviewers' comments.

We cannot make any decision about publication until we have seen the revised manuscript and your response to the reviewers' comments. Your revised manuscript is also likely to be sent to reviewers for further evaluation.

Sincerely,

Alison Marsden

Associate Editor

PLOS Computational Biology

Daniel Beard

Deputy Editor

PLOS Computational Biology

Reviewer's Responses to Questions

**Comments to the Authors:**

Reviewer #1: This is an ambitious multiscale modeling effort of CTC margination and adhesion dynamics that may contribute to the cancer pathophysiology and its therapy. It shows that the RBC hematocrit and blood flow shear rate play key roles in the CTC margination and adhesion behaviors. It is a well-written manuscript that sheds important light on overlooked vascular flow mechanisms that contribute to the tumor metastasis in early stage, some concerns regarding several technical aspects that may affect the simulation results, their validation, and their limitations need to be addressed and/or clarified by the authors.

1. Adhesion of circulating tumor cells (CTCs) to the microvessel wall largely depends on the blood hydrodynamic conditions, one of which is the blood viscosity. I am wondering whether the authors consider the blood viscosity in their simulations. If yes, can the blood viscosity parameters at different hematocrit compare to experimental values?

2. In this study, the simulation results suggest that a moderate hematocrit promotes CTC adhesion, while a relatively low or relatively high hematocrit can help to prevent the adhesion. However, in a previous simulation study (Xiao et al., Biomech Model Mechanobiol, 2017, 16:597-610), it shows that the CTC has an increasing probability of adhesion with the hematocrit due to the increased wall-directed force in small microvessel. Can the authors give some comment on this discrepancy?

3. In the model setup, the authors state there are 29704 ligand sites on the wall, 1176 receptor sites on the CTC's membrane. Why do they choose these values? Can it compare to experimental data? Also, in general, the experimental measured diameters of CTCs are about 10 ~ 20 microns, which is larger than the CTC diameter used in this study. Why do they choose the parameter of CTC diameter in this way?

4. In the Section of “Margination of a CTC in a straight tube”, the authors mentioned that the average shear rate is 272 s-1, while in the Section of “Adhesion of a CTC on a plate”, the shear rate is about 7000 s-1. Why do they choose two quite different values in the simulations? Also, in the section of “Adhesion of a CTC on a plate”, they only simulated the adhesion behavior of a deformable capsule. Since they did not study the adhesion of CTC, I suggest to change the title of this section.

5. There are some available work have focused on the modeling of CTC adhesion. Two of them are directly relevant the problem at hand. The authors should reference and comment them to show the novelty of their work.

1. Xiao et al. (2017), Effects of flowing RBCs on adhesion of a circulating tumor cell in microvessels, Biomech Model Mechanobiol, 16:597-610.

2. Dabagh et al. (2020), Localization of rolling and firm-adhesive interactions between circulating tumor cells and the microvasculature wall. Cell Mol Bioeng, 13:141-154.

Reviewer #2: In the paper, the authors performed three-dimensional (3D) simulations on the

behaviors of the tumor cells in microvascular network and predict the possible CTC adhesion locations. This manuscript exhibits some weaknesses that should be addressed before it can be considered for publication,

The size of the CTC is selected to be spherical with the diameter of 9 micrometer, authors need to provide reference for this size selection, what is the typical size range of CTC in circulation and how does the cell size affect margination, adhesion and the eventual the metastasis.

Again 1176 receptor sites are assigned to the CTC's membrane on what basis and any references for this selections

More important, as showed in the Table 1, the bending and shear stiffness of CTC are smaller than those of red cells, which is unreasonable assumption.

Details about the adhesion model should be moved to Method section and equations should be provided so that the readers can appreciate the meaning of the all parameters. Also, are those adhesion parameters applicable for modeling CTC, why not use an adhesion model designed specifically for CTC such as the one mentioned by the authors in the introduction section “A stochastic adhesive model was developed by Hammer and Apte [26], which has been widely used to theoretically and numerically investigate the adhesion of various cells, certainly including the CTCs.”

It is not clear the adhesion of the CTC at the bifurcation is permanent or temporary, if the CTC is flowing into a vessel with a size smaller than the CTC, could it block the vessel?

Authors only test three cases of hematocrit 20% 30% and 40% to draw the conclusion that CTC adhesion is more prominent at moderate hematocrit. This conclusion is not convincing. More cases of hematocrits should be examined considering the variation of hematocrit in the microcirculation is wide.

The key findings of this work is “These results suggest that the tumor cells may be more likely to extravasate at the microvascular bifurcations if the blood flow is slow and the hematocrit is moderate.” Could the authors provide any clinical evidence to back up or correlate with this finding. This should be added to the discussion section which is basically repeating the result section in the current form.

Also the authors should could review some literature on inhibiting of tumor metastasis through predicting the location where tumor cells extravasate from the circulation to strengthen the clinical significance of this work.

Reviewer #3: Our understanding of cancer metastasis is directly tied to knowing how and why CTCs will adhere and marginate. The work here is targeting these issues using a 3D computational model. They confirm a higher likelihood of adherence at bifurcations and investigate the influence of nearby RBCs. The authors suggest that CTCs are more likely to extravasate at bifurcations if the flow is slow and hematocrit moderate. They suggest the work provides near insights into cancer pathophysiology and its diagnosis and therapy. Overall the topic of the paper is interesting but it seems to need further data to really be able to make statistically significant conclusions from the experiments proposed.

The authors rightly point out that the origin of such studies comes from leukocyte literature. They mention on line 64 that the model from Hammer and Apte has been used for CTC models but a citation would strengthen this. When discussing prior work, Au et al in PNAS, Takeishi’s CTC model, Dabagh et al’s adhesion model, and then the work from Hynes and Pepona in CMBE and Science Advances show simulations in complex microvascular networks. Particularly the latter explore the idea of CTC adhesion at bifurcations. Differentiating from these works is important.

Why was a rat mesentery geometry chosen? Is it representative of a human’s? How does the ratio of cell size to vessel diameter compare? If we’re to make conclusions about human CTCs based on this, further justification/discussion is needed. Further, for the geometry, how was the geometry obtained and segmented? This work seems to lack details needed to understand the method. What was the adjustment made to achieve one outlet? Why was this needed? Are the CTCs introduced so that they don’t interact? The number of CTCs in this simulation as shown in Fig 1B seems unphysiologically high. It looks like some of the branches are nearing a diameter of 9 microns (ie similar or smaller to the CTC). If the branch size gets smaller than the CTC, can you really neglect the cell nucleus?

I’m confused about the first set of simulations. Typically when investigating cells in flow, you run the simulation through an initialization phrase so that the cells are found throughout the geometry. Then you can start studying the behavior of the CTCs. In this case, the authors are studying what happens as the geometry fills. This doesn’t seem realistic and could lead to behavior and interactions that are not representative of realistic cases. Why are measurements being made before some sort of convergence in Hct? They mention the CTCs moving away from the centerlines but it is difficult to quantify this here. This figure seems to support what has been shown often for leukocytes and was recently highlighted by Pepona et al, that CTCs will likely adhere at bifurcations. It’s good to confirm it but hard to identify the novel contribution in this particular study.

What are the effects of the initial RBC placement? It seems like fro Figure 4, this simulation is only completed once so it is hard to isolate the influence of the RBC placement. Showing the results under different instantiations would help discern the impact from the radial aggregation force. It’s great to be including results with the red blood cells vs just a CTC in the tube, but further information is really needed to sort out the impact of the RBC interactions. One of the conclusions relies on the CTC still being surrounded by RBCs at the end and suggests this is why you don’t see adhesion in a straight tube. This conclusion seems like a jump when looking at only one instantiation for a short period. If the simulation were to run longer (or in a longer tube) what would happen with the CTC near the wall? Would it continue to push toward the wall and make more contact or continue to be surrounded by RBCs? If the latter, why? There is also a discussion of shear rate in the margination section without simulation data for multiple shear rates, so it’s unclear how any conclusions about the impact of low shear can be drawn. From the methods section, it doesn’t look like there is an inter particle potential implemented? Interaction between individual RBCs and the RBC/CTC needs to be taken into account and can influence the CTC trajectory.

The adhesion work recreates the setup of Zhang et al. While the parameters are well described, there motivation seems lacking. Is it just because that is what Zhang et al did? Why is this representative of the question at hand? Much of this section is just recapitulated the prior results from [46]. The biggest difference seems to come from the bond number and it is suggested this is due to the deformation model used. Is a strain hardening or strain softening model more representative of the CTCs at hand? How should we interpret this change? It’s been shown in many previous papers that deformability, adhesion, and shear fluid influence adhesion. It’s hard to discern in this section what contribution is being added.

For the bifurcation work, they look at three different hematocrit levels. How were 20/30/40 chosen? I wonder if the 10% jump is too much? Will you see those levels in the microvasculature regions being looked at? Is this physiologically relevant? Again, it seems like only one simulation is completed so it’s hard to tell how much the initial position and orientation of the red blood cells influence. From (a) of Figure 8, it looks like you have artifacts from the setup phase before the CTC even hits the bifurcation which could influence it’s interaction. Typically we would see the simulation run to a steady state in terms of deformation of the nearby cells before measuring the effect on a CTC of interest. It’s difficult to make any serious conclusions from the singly simulation without getting past these initialization artifacts. Moreover, what more is added over the recent work from Pepona et al. in CMBE?

The author summary could be edited for clarity/English in a few places. For example "Thus, what conditions do the tumor cells 21 happen metastasis easily, and where do they extravasate from easily?” needs editing.

**Have all data underlying the figures and results presented in the manuscript been provided?**

Reviewer #1: Yes

Reviewer #2: Yes

Reviewer #3: Yes

PLOS authors have the option to publish the peer review history of their article (what does this mean?). If published, this will include your full peer review and any attached files.

Reviewer #1: No

Reviewer #2: No

Reviewer #3: No
---

## [Decision Letter · Decision Letter 1]

27 Jan 2021

Dear Dr. Ye,

We are pleased to inform you that your manuscript 'Margination and adhesion dynamics of tumor cells in a real microvascular network' has been provisionally accepted for publication in PLOS Computational Biology.

Best regards,

Alison Marsden

Associate Editor

PLOS Computational Biology

Daniel Beard

Deputy Editor

PLOS Computational Biology

Reviewer's Responses to Questions

**Comments to the Authors:**

Reviewer #1: I am satisfied with the way in which the authors have addressed my comments and recommend accepting the paper as is.

Reviewer #2: The authors have properly addressed all my concerns.

**Have all data underlying the figures and results presented in the manuscript been provided?**

Reviewer #1: Yes

Reviewer #2: Yes

PLOS authors have the option to publish the peer review history of their article (what does this mean?). If published, this will include your full peer review and any attached files.

Reviewer #1: No

Reviewer #2: No

---

## [Editor Report · Acceptance letter]

15 Feb 2021

PCOMPBIOL-D-20-01781R1 

Margination and adhesion dynamics of tumor cells in a real microvascular network

Dear Dr Ye,

I am pleased to inform you that your manuscript has been formally accepted for publication in PLOS Computational Biology. Your manuscript is now with our production department and you will be notified of the publication date in due course.

With kind regards,

Alice Ellingham
